# Porous Membranes of Polysulfone and Graphene Oxide Nanohybrids for Vanadium Redox Flow Battery

**DOI:** 10.3390/polym14245405

**Published:** 2022-12-09

**Authors:** Chien-Hong Lin, Ming-Yen Chien, Yi-Cih Chuang, Chao-Chi Lai, Yi-Ming Sun, Ting-Yu Liu

**Affiliations:** 1Chemistry Division, Institute of Nuclear Energy Research, Taoyuan City 32546, Taiwan; 2Department of Materials Engineering, Ming Chi University of Technology, New Taipei City 243303, Taiwan; 3Department of Chemical Engineering and Materials Science, Yuan Ze University, Taoyuan City 32003, Taiwan; 4Research Center for Intelligent Medical Devices, Center for Plasma and Thin Film Technologies, Ming Chi University of Technology, New Taipei City 243303, Taiwan

**Keywords:** vanadium redox flow battery, polysulfone, graphene oxide nanosheets, porous membranes, energy efficiency, green energy storage

## Abstract

Porous nanohybrid membranes of polysulfone (PSF) with graphene oxide (GO) nanosheets (PSF/GO membrane) were developed to serve as proton exchange membranes in a vanadium redox flow battery (VRFB). Various ratios of PSF/GO and thickness were investigated to evaluate the optimal voltage efficiency (VE), coulombic efficiency (CE), and energy efficiency (EE) of the VRFB. The pore size, distribution, and hydrophilicity of PSF/GO membranes were studied using scanning electron microscopy (SEM) images and contact angles. Functional groups of GO were evaluated using Raman spectroscopy. The mechanical properties and thermal stability of PSF/GO membranes were analyzed using a tensile tester and thermogravimetric analysis (TGA), respectively. The results show that the mechanical properties of the PSF porous membrane with GO nanosheets were significantly improved, indicating that the addition of graphene oxide nanosheets consolidated the internal structure of the PSF membrane. Cyclic voltammetry revealed an obviously different curve after the addition of GO nanosheets. The CE of the VRFB in the PSF/GO membrane was significantly higher than that in the pristine PSF membrane, increasing from 80% to 95% at 0.6 wt.% GO addition. Moreover, PSF/GO membranes displayed great chemical stability during long-term operation; thus, they can evolve as potential porous membranes for application in VRFBs for green energy storage.

## 1. Introduction

The advancement of modern industrial science and technology has accelerated the consumption of nonrenewable energy such as fossil fuels and coal, which has prompted the application of new energy to receive more and more attention around the world [1,2,3]. Renewable energy, including solar energy, wind energy, wave energy, and tidal energy, has the advantages of being inexhaustible and environmentally friendly. However, its intermittent and fluctuating nature leads to the difficulties of peak regulation and grid connection, which determine that its large-scale development must be supported by advanced energy storage technology [4]. The application of energy storage technology is very wide, providing stored power to mobile phones, electronic goods, game consoles, electric vehicles, and even industrial uninterruptible power systems (UPSs). The energy storage system is one of the key factors for the development of renewable energy. Because the energy storage system can adjust power, reserve capacity, and instantaneous power supply, it can reduce the impact of instantaneous changes in the power generation system on the power grid and is an important part of the green energy industry chain. The common energy storage devices in the world can be roughly divided into two categories: physical energy storage and chemical energy storage [5,6]. Chemical energy storage devices include batteries (lead–acid, lithium-ion, sodium–sulfur, and flow batteries) and hydrogen fuel cells. Physical energy storage devices include compressed air, pumped water storage, supercapacitors, and flywheels. Various energy storage methods have their advantages and disadvantages, but low-cost, high-energy density, and long cycle life are necessary criteria of the system [7,8,9].

The vanadium redox flow battery (VRFB) has the characteristics of high efficiency, long life, simple maintenance, large charge/discharge capacity, and the use of vanadium electrolyte to solve the fatal problem of traditional lead–acid batteries with regard to environmental pollution. The vanadium redox flow battery is safe to operate and has the advantage of infinite service life in theory; hence, the whole industry has entered the initial stage of marketization [10]. Since it was proposed by Skyllas-Kazacos of the University of New South Wales in 1985, it has attracted the attention of many researchers and industries. Compared with lead–acid batteries and nickel–metal hydride batteries on the current market, vanadium flow batteries have obvious technical advantages such as high power, long life, safety, support for deep charge and discharge, and being green and pollution-free [11].

The VRFB is a kind of redox renewable battery energy storage system based on the vanadium element. Vanadium batteries store electrons in a sulfuric acid electrolyte of vanadium ions of different valences by means of chemical energy storage. The electrolyte is injected into the battery module through an external pump, and it circulates in different storage tanks and closed circuits under the action of mechanical power [12,13]. A porous membrane is used in the middle of the double-electrolyte tank as the separator of the battery pack. The electrolyte flows through the surface of the electrode in parallel and undergoes an electrochemical reaction. The current is collected and conducted through the carbon electrode plate, such that the chemical energy stored in the solution is converted into electrical energy. After the reaction, the solution returns to the storage tank, and the above steps are used to continuously complete the charge/discharge reaction. If many single vanadium flow batteries are connected in series, a high-voltage, high-capacity vanadium flow battery can be assembled. If it is matched with renewable energy sources such as photovoltaics and wind power that need to be adjusted in off-peak periods and store electric energy with high efficiency, it can help to improve the stability of renewable energy to the power supply.

Vanadium is a readily available material and is often used in steelmaking and chemical catalysts. It is naturally occurring and can also be recycled from various wastes. This research is dedicated to improving the performance of vanadium redox flow batteries, simplifying the production procedures and reducing production costs to achieve the goal of battery commercialization as soon as possible. From the perspective of research and development, the most critical aspect for flow battery technology at present is the development of breakthrough materials. In this study, we focus on the research on the modification of the key material, the porous membrane. The porous membrane is a polymer membrane that can conduct protons. The traditional understanding in research is that the material needs to adsorb a certain amount of electrolyte and can drive protons effectively after adding a fixed external current [14,15,16]. The biggest problem with the current use of proton exchange membranes (PEMs) for direct flow batteries is that higher voltages and electrolytes are highly oxidizing toward the battery electrodes, whereby the materials used in the porous membranes and fluid handling components require more chemical resistance. Reducing the cross-contamination of vanadium ions across the exchange membrane is also considered as a challenge, as is the use of a rather expensive ion exchange membrane (such as the Nafion membrane made by DuPont). Therefore, the development of low-cost, highly efficient membranes is the main purpose in this study [17,18,19]. The polysulfone (PSF) and graphene oxide (GO) nanohybrid membranes (PSF/GO membranes) are demonstrated as a novel PEM for VRFB to improve its mechanical properties, as well as thermal and chemical stability, and enhance the voltage efficiency (VE), coulombic efficiency (CE), and energy efficiency (EE).

## 2. Materials and Methods

### 2.1. Preparation of Graphene Oxide

First, fluffy graphite powder (1 g) was poured into 36 mL of H_2_SO_4_ in a container, and then the solution was stirred for 40 min. In the ice bath state, 12 mL of H_2_NO_3_ (concentration 100%) was slowly added dropwise, followed by 5 g of KMNO_4_, and the mixture was stirred at room temperature for 50 min. In an ice bath, 120 mL of DI water was slowly added, and stirring was continued for 1.5 h. After adding 6 mL of H_2_O_2_ solution, stirring was continued for 2 h, and then the mixture was allowed to stand for 6 h. Subsequently, the upper layer of the clear solution was removed, 200 mL of DI water was added, and then 1 mL of H_2_O_2_ solution and 1 mL of HCL were added before stirring for 1 h. The resulting samples were centrifuged and washed 2–3 times. Finally, the samples were washed with DI water, centrifuged 5–6 times, and then frozen and vacuum-dried (−50 °C) for storage.

### 2.2. Preparation of Pristine PSF and PSF/GO Nanocomposite Membrane

The pristine PSF or PSF/GO nanocomposite membrane was fabricated by the phase inversion technique using a film blade-coater. The PSF polymer powders (6.44 g) with DMF solvent (47.2 g) and GO nanosheets were completely dissolved in different weight percentages (0–1 wt.%), and then left to stand still for 3 days. The mixture was stirred evenly with a glass rod evenly and then dispersed using ultrasonic processors to suspend the GO nanosheets. The mixture is dropped onto a film blade-coater and scraped evenly across the glass substrate. The resulting pristine PSF or PSF/GO membrane was immersed in 10 L of DI water for 1 day to remove DMF solvent from the pristine PSF or PSF/GO membrane. The DMF would be dissolved with DI water and depart from the membrane, and then the pristine PSF or PSF/GO membrane would be precipitated in the DI water. Finally, the membrane was dried at room temperature. The wt.% of PSF membrane is shown in the following equation.
$$\frac{\mathrm{PSF}\left(g\right)}{\mathrm{PSF}\left(g\right)+\mathrm{DMF}\left(g\right)}\times 100\%=x\;\mathrm{wt}.\%$$

### 2.3. Material Characterization

The morphology and the pore volume and pore-size distribution of membranes were characterized using a scanning electron microscope (SEM) according to the Barrett, Joyner, and Halenda (BJH) analysis. The samples were coated with platinum before the measurement to improve conductivity, enabling a clear SEM photo to be taken. To determine pore volume and pore size distribution, the gas pressure was gradually increased until all pores were filled with liquid. Next, the gas pressure was gradually reduced to evaporate the condensed gas from the system. The evaluation of adsorption and desorption isotherms revealed information about the pore volume and pore size distribution. BJH calculations were used to determine the pore volume and pore size distribution. The oxidation of GO in the polysulfone membrane was characterized by Raman spectroscopy. Furthermore, the phase identification of graphite and GO was achieved using X-ray diffraction (XRD, Malvern Panalytical, Malvern, England) in a powder module, with CuKα radiation at a current of 40 mA and voltage of 45 kV. The membranes were cut into 40 × 20 mm pieces and clamped in the tensile equipment. 

### 2.4. Proton Conductivity

The proton conductivity of the composite membranes was measured using impedance spectroscopy (EIS) with a Autolab potentiostat/galvanostat (Ecochemie, Meterhm Autolab, Utrecht, Netherlands). The tests were carried out at a signal amplitude of 50 mV in the frequency range of 100 Hz–1 MHz. Furthermore, the proton conductivity of the composite membranes was calculated from the impedance data using the following equation:$$\sigma =\frac{L}{R\times S},$$
where *σ*, *L*, *R*, and *S* are the proton conductivity (S/cm), membrane thickness (cm), resistance from the impedance data (Ω), and cross-sectional area (cm^2^) of the membrane, respectively. The value of *R* was obtained from the intersection at high frequencies.

### 2.5. Permeability of Vanadium Ions

The permeability of vanadium ions of the pristine PSF membrane and modified PSF membrane were measured using a diffusion tank. The left reservoir contained 1.2 M VO^2+^ in 2.5 M H_2_SO_4_/3 M HCl, whereas the right reservoir contained 1.2 M MgSO_4_ in 2.5 M H2SO_4_/3 M HCl. The VO^2+^ solution was prepared by dissolving 1.2 M VOSO_4_ in 2.5 M H2SO_4_/3 M HCl solution. The vanadium ion permeability was calculated as follows [19,20]:$$V_{R}\frac{dC_{R}\left(t\right)}{dt}=\frac{AP}{L}\left[C_{L}-C_{R}\left(t\right)\right]$$
where *C_L_* is the vanadium ion concentration in the left reservoir; *C_R_(t)* is the vanadium ion concentration in the right reservoir as a function of time; *A* and *L* are the area and thickness of the membrane, respectively; *P* is the permeability of the vanadium ions; and *V_R_* is the volume of the right reservoir. It is supposed that the change in vanadium ion concentration in the left reservoir is always negligible, and a pseudo-steady-state condition is achieved inside the membrane. An assumption is also made that *P* is independent of the concentration [8,20]. Therefore, the following equation can be obtained for calculating the permeability:$$P=\frac{LV_{R}}{AC_{L}}\frac{C_{R}\left(t\right)}{t}$$

The concentration of VO^2+^ in the right reservoir was titrated with KMnO_4_ using an electrometric titrator (877 Titrino plus, Metrohm USA Inc., Riverview, FL, USA).

### 2.6. VRFB Single-Cell Test

For the VRFB single cell used in the charge/discharge tests, the solutions of 1.7 M V^2+^/V^3+^ in 3.5 M H_2_SO_4_ and 1.7 M VO^2+^/VO_2_^+^ in 3.5 M H_2_SO_4_ were employed as the cathode and anode solutions, respectively. The carbon felts served as the electrodes and the graphite plates were the current collectors. The active area of the electrode was 25 cm^2^, while the volume of the solution in each half-cell was 80 mL. To avoid the corrosion of the carbon felt electrode and graphite plates, the VRFB was charged to 1.6 V and discharged to 0.7 V. The cycling life test was operated under a current density of 120 mA·cm^−2^. The coulombic efficiency (CE), voltage efficiency (VE), and energy efficiency (EE) of the cell were calculated in the following equation. CE = Discharge capacity / Charge capacity; EE = Discharge power / Charge power; VE = EE/CE.

## 3. Results and Discussion

### 3.1. Morphology

The porous PSF membrane was prepared using the phase inversion technique, which could form microscale pores, as observed by SEM images (Figure 1). The surface pore size of PSF porous PEMs gradually increased with the decrease of the PSF concentration. Furthermore, the pore size distribution of the PSF membrane was 1.14 µm at 20 wt.%, 1.80 µm at 16 wt.%, and 3.49 µm at 12 wt.% PSF, as calculated using Image J software (Figure 2). Although microscale pores were observed in SEM images, there were many nanoscale channels and pores inside the porous membrane, unlike the surface structures. The surface pore size of 16 wt.% PSF was about 1.80 µm according to SEM analysis, but the real inner diameter of the nano-channel was only 30 µm (Table 1), as measured by the pore size analyzer from the Textile Research Center, Taiwan. The gas adsorption experiments are used to characterize the surface area, pore size distribution, and pore volume of those pores accessible from the surface of porous materials. A wide range of pore sizes from 0.35 nm to over 100 nm can be analyzed with high accuracy using volumetric or gravimetric adsorption techniques [21].

In principle, the pore size affects the coulombic efficiency (CE). A smaller pore size induced a higher CE; for example, the CE of 20 wt.% PSF was higher than that of the 12 wt.% and 16 wt.% PSF membranes. The reason is that the larger pore size increased the faster flow rate of vanadium ions, enabling the electrolyte to directly pass through the porous membrane and cause pollution, which reduced the CE [22,23]. However, compared with 16 wt.% PSF (inner pore size: 30 nm), the vanadium ions struggled to pass through the ultrasmall inner pore size (10 nm) of 20 wt.% PSF, thereby decreasing voltage efficiency (VE). Accordingly, 16 wt.% PSF membranes were used as the model membranes for further GO nanosheet addition, due to the best energy efficiency (EE = CE × VE) obtained as a function of the optimal pore size. The issue of charge/discharge efficiency is discussed in Section 3.7.

The cross-section SEM images of the 16 wt.% pristine PSF and 16 wt.% PSF/0.6 wt.% GO membranes are shown in Figure 3a,d, revealing a finger-like structure. Furthermore, the thickness increased from 140 to 300 μm after adding GO nanosheets, thus enhancing the CE of the membranes. Furthermore, the addition of GO nanosheets also increased the VE of the membranes, owing to the existence of the carboxyl group of GO and its higher conductivity [24]. Moreover, the top-view SEM images (Figure 3b,e) reveal that the surface pore size was larger after blending with GO nanosheets, but the inner pore size of membranes was not affected. Figure 3c,f indicates the optical photos of pristine PSF and PSF/GO membranes, respectively. The homogenous dispersion of GO nanosheets (brown color) into PSF membrane were found in in Figure 3f.

### 3.2. Raman Spectroscopy

Raman spectra of the porous membrane after doping graphene oxide with different concentrations are shown in Figure 4. The PSF/GO modified membrane shows fundamental peaks in 1250 to 1750 cm^−1^ region. Raman spectra of 0.4 wt.% doping amount of GO nanosheets were found the peak at ~1380 cm^−1^ [25] and can be associated with the D vibration band caused by the disordered structure of GO, suggesting a small and broad peak from graphene oxide nanosheets [26]. Meanwhile, the G vibration band at 1590 cm^−1^ correspond to sp2 carbon atoms plane stretching motion [27]. When the doping amount of GO nanosheets reaches 0.6 wt.%, the obvious D and G bands begin to appear, and the full characteristic peaks of GO can be observed in samples with 0.8 and 1 wt.% GO, which confirms that GO nanosheets have been successfully blended in PSF membrane.

### 3.3. Tensile Strength

The mechanical properties of the porous PSF membranes were significantly improved as a function of different concentrations of GO nanosheets (Figure 5). The optimal strain was found with the 0.4 wt.% addition of GO nanosheets, highlighting an obvious sixfold increase compared with the pristine PSF membrane. The strain decreased from 60% to 40% following the addition of 0.6 wt.% GO nanosheets. However, the increase in stress caused the strain to gradually decrease for the addition of GO nanosheets up to 0.8 wt.%, indicating that a higher content of GO nanosheets led to a more brittle structure. The improvement of both tensile strength and elongation observed for the PSF with 0.6 wt% GO content could be explained considering that the incorporation of GO nanosheets into the PSF matrix can enhance the mechanical strength of the composite membrane owing to the hydrogen-bonding interaction between sulfonated acid groups of GO nanosheets and oxygen-containing groups of the PSF matrix [28,29].

### 3.4. Thermogravimetric Analysis (TGA)

Figure 6 shows the results of TGA analysis of PSF membranes. The first visible weight loss of PSF porous membrane occurs from 100 to 230 °C, and can be attributed to the decomposition of sulfonic group [30]. The other PSF/GO porous membrane could still maintain thermal stability due to the well dispersed GO nanosheets within the PSF matrix, which could obviously promote the thermal stability by controlling the polymer chain mobility [31]. The pristine PSF porous membranes have an excellent initial degradation temperature (T_i_) from 496 to 523 °C. Since the PSF membranes composited with different proportions of graphene oxide, the porous membranes maintained a stable T_i_ during the measurement. As the temperature increased, the PSF porous membranes were not entirely degraded, especially the 0.6 wt.% membrane, which left around 38% residue at 700 °C and was obviously higher compared to the pristine PSF membrane (30%). The outstanding thermal stability could be attributed to the intercalation of the GO matrix into the exchange tunnels, playing the role of a stable wall for the thermal degradation of polysulfone. The results reveal that the thermal stability of PSF can exhibit optimal performance in high temperature working environments.

### 3.5. Permeability of Vanadium Ions

The membrane needs to separate the electrolytes in each half-cell to restrict the cross-mixing of vanadium ions; thus, the rate of cross-mixed vanadium ions was characterized by vanadium permeability [22,23]. To obtain a high coulombic efficiency and low self-discharge rate, the membrane used in the VRFB system should have a low permeability of vanadium ions. Figure 7 shows the changes in the concentration of VO^2+^ in the right reservoir with time due to transfer across the membrane. The results show that the permeation rate of vanadium ions through the PSF membrane was much higher than that through the PSF/GO membrane under the same conditions. The permeability of VO^2+^ across the pristine PSF membrane was 25.9 × 10^−7^ cm^2^·min^−1^, while that across the PSF/GO membrane was 12.3 × 10^−7^ cm^2^·min^−1^, i.e., only 56% of the previous value. The decrease in the permeability of vanadium ions through the modified membrane could be attributed to the blocking of the pores on the PSF membrane surface by GO. In addition, the PSF/GO membrane has smaller hydrophilic/hydrophobic separation and higher dispersed -COOH groups, causing the narrow and branched water filled channels. Thus, the migration of vanadium ion in PSF/GO membranes is slower. Moreover, the addition of 0.6 wt% GO would further separate the hydrophilic domain in the hybrid membranes, which increases the difficulty of vanadium ion permeation, leading to a lower and good trend of VO^2+^ permeability [29,31].

### 3.6. Electrochemical Impedance Spectroscopy (EIS)

EIS can be used to analyze the interior resistance of the VRFB single cell. When the resistance Z″ (Ω) was at 0, the value of resistance Z′ (Ω) is the real interior resistance of cell (Figure 8). The results reveal that the introduction of GO into the PSF membrane significantly decreased the interior resistance, indirectly indicating that the hydrophilic characteristics of GO led to the greater absorption of vanadium electrolyte into the cell, thus decreasing the interior resistance of the battery. When the amount of GO was increased to 1 wt.%, the resistance worsened due to the low conductivity of GO. As a result, the 0.6 wt.% GO membrane had the lowest interior resistance, facilitating the convenient charging of the VRFB. In order to improve the electrochemical reaction by GO, Cui et al. [30] reported the preparation of graphene oxide (GO) nanosheets (GONPs) using an improved Hummers’ process. Compared with the original graphene, due to the introduction of oxygen-containing groups, GONPs show a good catalytic ability for the V^2+^/V^3+^ redox reaction. The samples processed under 50 °C and vacuum environment especially have the best catalytic performance for VO^2+^/VO_2_^+^ and V^2+^/V^3+^ redox reactions. This is because the content of hydroxyl and carboxyl functional groups is the highest currently. Blanco et al. [32] obtained thermally-reduced graphene oxide (TRGO) at 1000 °C and inert atmosphere. The material was applied to the positive electrode reaction of VRFB. TRGO shows high conductivity and low overpotential for VO^2+^/VO_2_^+^ redox reactions. High VE and EE were obtained in the charge/discharge experiment [33].

### 3.7. Charge/Discharge Efficiency

EE is a very important indicator because it reflects the energy conversion efficiency of large-scale energy storage devices. The change in EE is dependent on the CE and VE [28]. The membrane produced (Figure 9) with 16 wt.% doping exhibited an insufficient CE, but its VE surpassed that of the porous membranes (Nafion) on the market. However, the overall EE was unable to surpass the porous membranes on the market. In the long-term test, we found that it exhibited the same properties as the exchange membranes on the market, which could be maintained for about 200 cycles. With the exception of the 12 wt.% PSF membrane, this was due to the pore size of the porous membrane being too large, resulting in the incomplete reaction of the vanadium electrolyte. The valence of the positive electrolyte was not converted (from VO^2+^ to VO_2_^+^); thus, transfer to the negative electrolyte to induce normal charging and discharging could not be performed after six cycles. The EE and CE of the 12 wt.% and 20 wt.% PSF porous membranes did not surpass those of the current porous membranes on the market as expected, but their VE was superior. Specifically, it was found that the 20 wt.% PSF porous membrane had the best conductivity, leading to a high CE, but a low charge/discharge efficiency. This led to a low VE, which indirectly affected the overall EE, compared with the 16 wt.% PSF membrane.

The 16 wt.% PSF porous membrane with 0.4 wt.% or 0.6 wt.% GO (Figure 9d,e) could maintain the CE at 96% during the long-term test. This is because the composition of GO nanosheets not only improved the mechanical properties, but also stabilized the pores inside the porous membrane, enabling the vanadium electrolyte to remain in the structure for a long time. A higher CE indicates less crossover of vanadium ions. In the PSF + 0.6 wt.% GO membrane, the pores in the PSF membrane were blocked by the GO nanosheets, thereby reducing the VO^2+^ permeability. 

Although the VE was not significantly improved, it was maintained at a certain level, indirectly increasing the EE. The VE was maintained at about 70% for 0.4 wt.% and 0.6 wt.% GO addition, whereas the CE at the 0.6 wt.% GO/PSF membrane was better than that of the 0.4 wt.% GO/PSF membrane, indirectly proving that the structure of the former was more stable. The addition of GO may increase the tortuosity of permeation pathway for vanadium ions; therefore, GO/PSF composite membrane may have lower permeability of vanadium ions to induce the higher CE. Furthermore, the carboxylated groups on GO may provide some proton exchange capability and increase the water channels for proton transport in GO/PSF membranes, which can enhance the VE. However, the 1 wt.% GO/PSF porous membrane could not be charged and discharged because the pores were filled with GO nanosheets. The 0.6 wt.% GO/PSF membrane exhibited a higher EE (i.e., the product of CE and VE), as both the CE and the VE of the PSF/GO porous nanohybrid membranes were higher than those of the pristine PSF membrane.

Besides ion exchange membranes (IEMs) such as cation exchange membranes (CEMs), anion exchange membranes (AEMs) and amphoteric ion exchange membranes (AIEMs), nonionic porous (micro or nanoporous) membranes have been developed for application in VRFBs. The porous membranes obstruct vanadium ion transport, but allow proton transfer via pore size exclusion effect. In comparison with the commercial IEMs such as Nafion membranes, porous membranes are of remarkably lower cost; thus, they may be excellent alternatives to potentially replace the highly expensive IEMs [34,35,36]. Che et al. [37], by altering the added amount of silica spheres in the membrane fabrication process and using 3 M NaOH as an etching solution, found that the porosity of the porous PBI membranes is effectively regulated. The VRFB assembled with the PBI-40%SiO_2_ membrane displays excellent CE% (99.5–100%) and EE% (87.9–71.5%) at high current densities of 80–200 mA cm^−2^, as well as superior cycling stability at 120 mA cm^−2^ in the vanadium flow battery. Mu et al. [38] fabricate a series of low-budget and high-performance blend membranes from polyvinylpyrrolidone (PVP) and cardo-poly(etherketone) (PEKC) for VFRB. The VRFB equipped with 50%PVP-PEKC membrane has high coulombic efficiencies (99.3–99.7%), voltage efficiencies (84.6–67.0%), and energy efficiencies (83.9–66.8%) at current densities of 80–180 mA cm^−2^. The cycling performance of the VRFB equipped with 50% PVPPEKC at 100 mA cm^−2^ was stable and maintained almost unchanged CE, VE, and EE during the entire cycles. GO and its derivatives were often used as fillers for polymer composites, and their composites are known to exhibit good electrochemical and mechanical properties. Similar to GO organic fillers, Pandit et al. suggest that the hierarchically organized α-MnO_2_ sufficiently satisfies the stringent requirements of electrode materials in constructing high-power and long-term life batteries for energy storage applications. Furthermore, we expect that novel efforts in finding newer electrolytes and their additives will overcome the present hurdles in achieving higher energy density with stable cycle performance for sodium-ion batteries [39]. In addition, these authors have synthesized hexagonal δ-MnO_2_ nanoplates via a facile chemical method. The electrode shows key electrochemical properties, including the excellent reversible capacity (108 mAh/g at C/20), improved rate capability performance, stable cyclic studies, and almost 100% coulombic efficiency for potassium-ion batteries [40].

In this study, GO nanosheets were uniformly filled in the PSF matrix forming a composite membrane denoted as PSF/GO. The unique two-dimensional structure and the -COOH functional groups of GO nanosheets make it compatible with PSF matrix, while the VRFB system could be improved. Table 2 summarizes the basic VRFB performance of pristine PSF and PSF/GO composite membranes. The vanadium ion permeability of PSF + 0.6 wt% GO membrane was significantly decreased 53% (from 25.9 to 12.3 × 10^−7^ cm^2^ min^−1^), compared with the pristine PSF membrane. However, the CE of PSF + 0.6 wt% GO membrane increases from 79.6 to 95.5%. The VRFB assembled with the PSF + 0.6 wt% GO membrane exhibited the highest cell performances; the EE of PSF + 0.6 wt% GO membrane exhibited a 6.3% increase compared with the pristine PSF membrane. The PSF + 0.6 wt% GO membrane has superior chemical and mechanical stability and can be repeatedly used in VRFB without any damage, as confirmed by a test of 300 cycles.

## 4. Conclusions

The commonly used separators in VFRB are perfluorosulfonic polymers, such as Nafion membranes, owing to their high proton conductivity and excellent chemical and electrochemical stability. However, the costs of Nafion series membranes for large-scale VRFB systems are still extremely high. The modified Nafion membranes are not suitable for the commercialization of VRFB systems. Therefore, the alternative low-cost membrane with high performance is imminently required for the wider usage of VRFB. Owing to the feature of their low cost, the nonionic porous membranes have been studied for VRFB applications. The porous membrane could separate vanadium ions from protons via pore size exclusion due to the differences in the stokes radius, causing low vanadium ion permeability and relatively high VRFB performance. PSF/GO porous membranes have a low cost, simple manufacturing process, and environmentally friendly materials.

Therefore, PSF/GO porous nanohybrid membranes were successfully fabricated using various ratios of PSF and GO nanosheets. The pore size decreased with an increase in the concentration of PSF. A smaller pore size led to a higher CE but reduced the VE. Thus, the optimal pore size was 16 wt.% PSF, showing an outstanding overall EE. After the addition of GO nanosheets, the mechanical properties were improved, the internal structure of the porous membrane was strengthened, and the conductivity was increased, thereby resulting in a greater EE. The Raman spectra exhibited the obvious characteristic peaks of GO nanosheets, confirming that they were homogenously blended into the PSF membrane. The CE could be maintained at 96% after doping with 0.6 wt.% GO nanosheets. Thus, the high stability of the 16 wt.% PSF + 0.6 wt.% GO membranes have potential application in VRFBs for green energy storage.

## Figures and Tables

**Figure 1 polymers-14-05405-f001:**
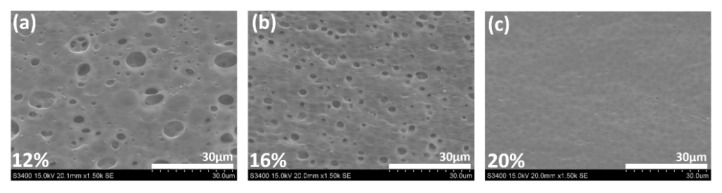
SEM images of porous PSF membranes: (**a**) 12 wt.%; (**b**) 16 wt.%; (**c**) 20 wt.% PSF in DMF.

**Figure 2 polymers-14-05405-f002:**
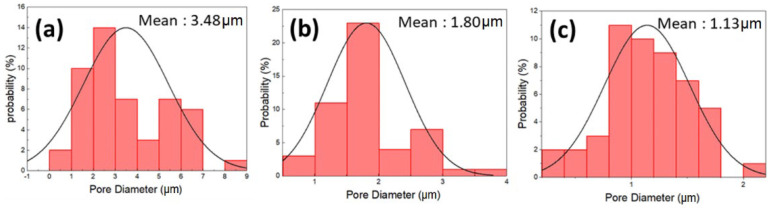
Pore size analysis of PSF membranes: (**a**) 12 wt.%; (**b**) 16 wt.%; (**c**) 20 wt.%.

**Figure 3 polymers-14-05405-f003:**
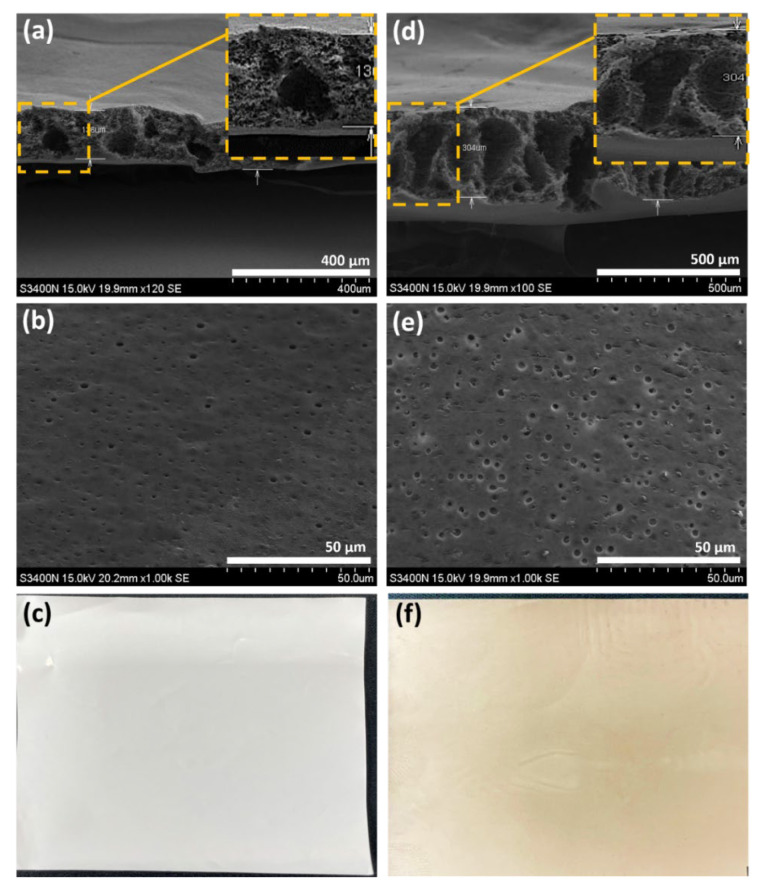
SEM images of (**a**) cross-section, (**b**) top view, (**c**) optical photo of 16 wt.% pristine PSF membrane; and (**d**) cross-section, (**e**) top view, (**f**) optical photo of 16 wt.% PSF/0.6 wt.% GO membrane.

**Figure 4 polymers-14-05405-f004:**
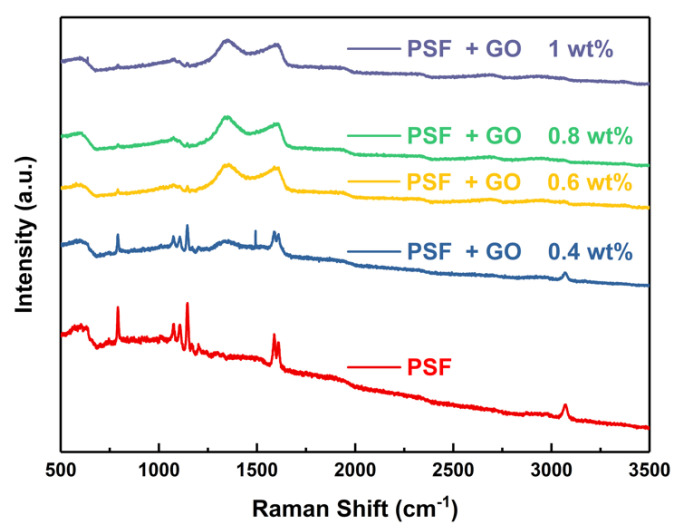
Raman spectra of PSF and PSF/GO membranes.

**Figure 5 polymers-14-05405-f005:**
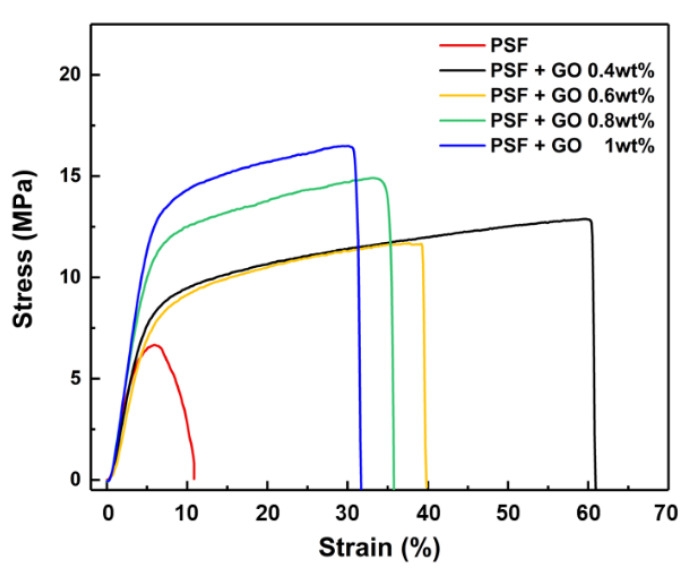
The tensile strength of PSF and PSF/GO membranes.

**Figure 6 polymers-14-05405-f006:**
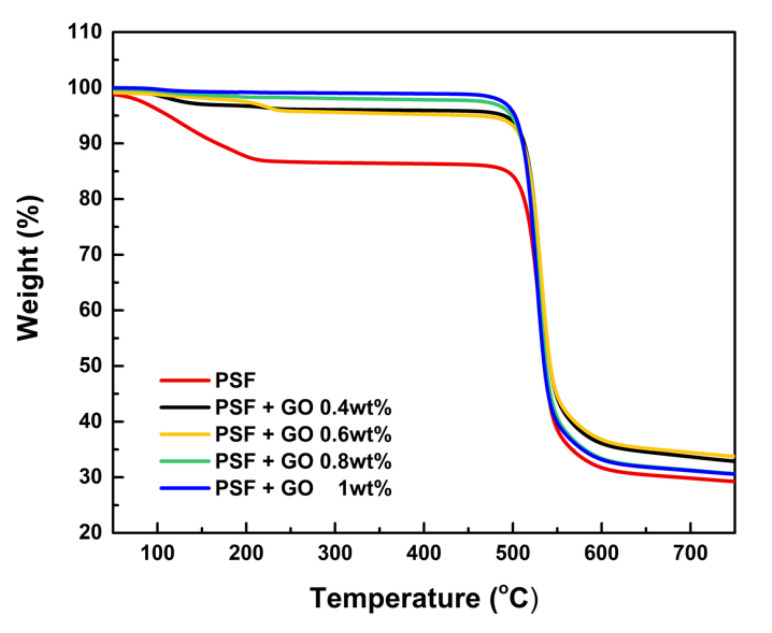
TGA analysis of pristine PSF and PSF/GO membranes.

**Figure 7 polymers-14-05405-f007:**
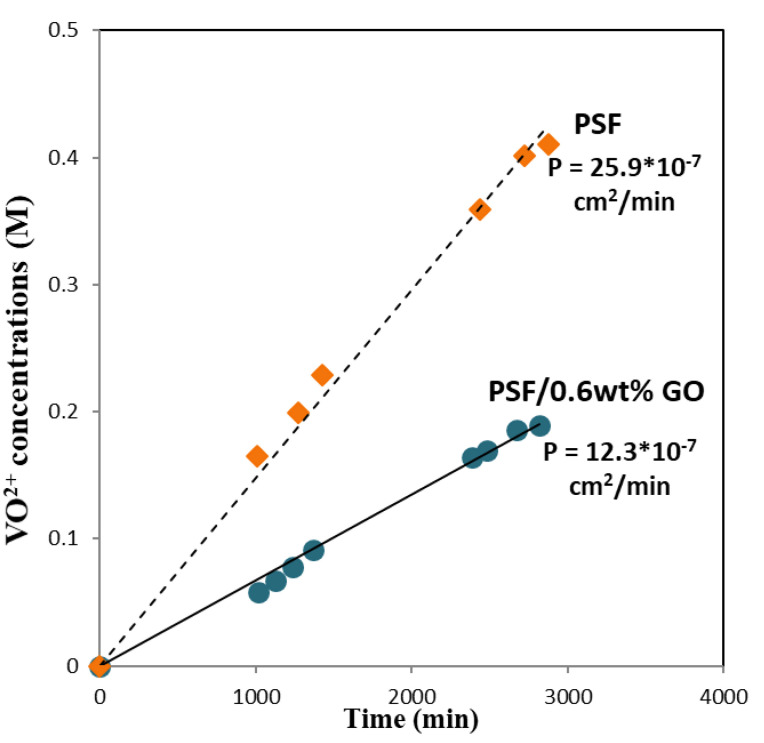
Vanadium ion permeability of PSF/GO membranes.

**Figure 8 polymers-14-05405-f008:**
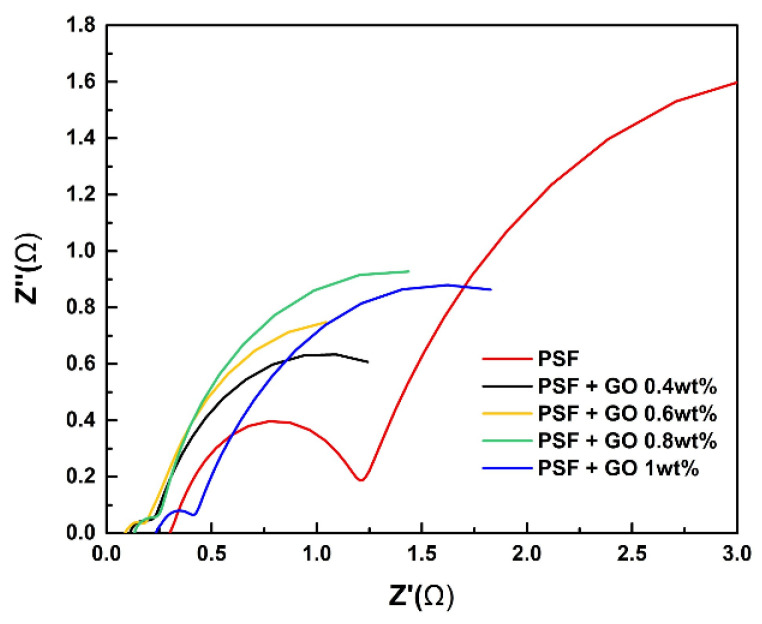
Nyquist plots of various PSF/GO membrames from EIS measurement.

**Figure 9 polymers-14-05405-f009:**
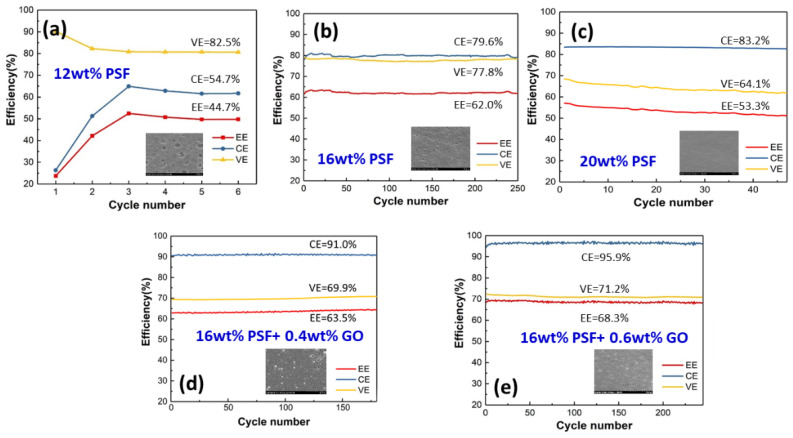
The charge/discharge efficiency of the (**a**) 12 wt.%, (**b**) 16 wt.%, and (**c**) 20 wt.% PSF membranes, and of the (**d**) 16 wt.% PSF + 0.4 wt.% GO and (**e**) 16 wt.% PSF + 0.6 wt.% GO membranes.

**Table 1 polymers-14-05405-t001:** Surface and inner pore size analysis at various weight percentages (wt.%) of polysulfone (PSF) membrane.

Membranes	Average Surface Pore Size according to SEM (nm)	Inner Pore Size according to Pore Size Analyzer (nm)
20 wt.% PSF	1139	10
16 wt.% PSF	1802	30
12 wt.% PSF	3487	470

**Table 2 polymers-14-05405-t002:** Characterizations comparison of pristine PSF and PSF/GO membranes.

	Pristine PSF	PSF + 0.6 wt% GO
Vanadium ion permeability (10^−7^ cm^2^ min^−1^)	25.9	12.3
Coulombic efficiency (CE) (%)	79.6	95.9
Voltage efficiency (VE) (%)	77.8	71.2
Energy efficiency (EE) (%)	62.0	68.3

## Data Availability

The data presented in this study are available on request from the corresponding author.

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
