# Peer review of "Porous Membranes of Polysulfone and Graphene Oxide Nanohybrids for Vanadium Redox Flow Battery"

_polymers, 2022, doi:10.3390/polym14245405_

Round 1

Reviewer 1 Report

The manuscript "Improving the performance of vanadium redox flow battery with porous membrane" develope porous nanohybrid membranes of polysulfone (PSF) with graphene oxide (GO) nanosheets (PSF /GO membrane) to serve as proton exchange membranes in vanadium redox flow battery (VRFB). The prepared PSF /GO  membranes were scientifically characterized and the results were discussed in detail. CE of VRFB with PSF/GO membrane is higher than that of pristine PSF membrane. Moreover, PSF/GO membranes display good chemical stability during long-term operation and thus evolve as potential porous membranes for application in VRFB for green energy storage. Following issues need be addressed before further consideration.

1.     The porous PSF membrane prepared by the phase inversion technique, please describe in detail how the PSF porous membrane is prepared by phase conversion technology and explain what x stands for in x%wt PSF membrane.

2.     The Raman spectrum in this paper lacks data support from other references.

3.     Please mark the scale ruler on the SEM image.

4.     There are no photos of the membranes. Please provide clear and regular photos of the pristine and composite membranes and match them with SEM and one by one.

5.     Please enlarge the scale of cross-section SEM images in order to ensure if the porous structure throughout the membrane.

6.     The vertical coordinate of the Thermo-gravimetric Analysis diagram should be weight%’

7.     Line 255-256 on page 7,The result reveals that the magnificent thermal stability of PSF can have optimum performance in high temperature working environments.’ This conclusion is not correct according to the TG data. Meanwhile, why does the pristine PSF membrane Thermo-Gravimetric curve drop at 100-200 degrees?

8.     Provide the clear data of other porous membranes (such as 10.1016/j.memsci.2020.118359) and blend membranes (10.3390/batteries8110230) in the vanadium flow cell compared with the data in this paper and then clarify the advantages of the PSF/GO membranes.

9.     Please provide more discussions on the reasons for higher mechanical strength and low vanadium ions permeability of PSF/GO composite membranes comparing with pure porous PSF membrane.

10. Provide equations to calculate the coulombic efficiency (CE%), energy efficiency (EE%) and voltage efficiency (VE%).

Reviewer 2 Report

The manuscript entitled “Improving the performance of vanadium redox flow battery 2

with porous membrane” has been submitted by the authors. Some issues to be addressed will improve the quality of the manuscript. Therefore, I recommend this work could be published after the major revision

1.      The author should write down the novelty of this paper.

2.      The English composition requires many improvements. The authors should proofread the manuscript carefully to minimize grammatical errors.

3.      All the references mentioned in the paper should be cited in the text or vice-versa.

4.      This research topic has been widely studied, and many studies have been performed. The author, please add a comparative table for the reader's clear understanding.

5.      Font size of Fig. 2 on both x and y-axis very small.

6.       In Fig. 3, the scale bar on the SEM measurement cannot be seen very well.

7.      Enough references do not support the characterization and result and discussion parts. It may be supported by the recent relevant references (before 2015).

ACS Appl. Mater. Interfaces 2021, 13, 9, 11433–11441; Ceramics International

Volume 48, Issue 19, Part B, 1 October 2022, Pages 28856-28863; Composites Part B: Engineering Volume 242, 1 August 2022, 110094

Round 2

Reviewer 1 Report

The authors have addressed all the suggested modifications and I can now accept the manuscript in its present form for publication.